# Application of Imaging Algorithms for Gas–Water Two-Phase Array Fiber Holdup Meters in Horizontal Wells

**DOI:** 10.3390/s24227285

**Published:** 2024-11-14

**Authors:** Ao Li, Haimin Guo, Yue Niu, Xin Lu, Yiran Zhang, Haoxun Liang, Yongtuo Sun, Yuqing Guo, Dudu Wang

**Affiliations:** 1College of Geophysics and Petroleum Resources, Yangtze University, Wuhan 430100, China; 2022730026@yangtzeu.edu.cn (A.L.); 2022710374@yangtzeu.edu.cn (H.L.); 2022720524@yangtzeu.edu.cn (Y.S.); 2023710415@yangtzeu.edu.cn (Y.G.); 2023720526@yangtzeu.edu.cn (D.W.); 2Key Laboratory of Exploration Technology for Oil and Gas Resources of Ministry of Education, Yangtze University, Wuhan 430100, China; 3Sinopec Shengli Oilfield Reservoir Dynamic Monitoring Center, Bingzhou 256600, China; ny401z@163.com; 4Jianghan Oil Production Plant, Jianghan Oilfield Branch, Sinopec Corp., Qianjiang 433124, China; luxin981210@163.com; 5PetroChina Dagang Oilfield Company, Tianjin 300280, China; zhangyiran117@foxmail.com

**Keywords:** production logging, gas array tool, horizontal well, gas–water two-phase flow, water holdup, imaging algorithm

## Abstract

The flow dynamics of low-yield horizontal wells demonstrate considerable complexity and unpredictability, chiefly attributable to the structural attributes of the wellbore and the interplay of gas–water two-phase flow. In horizontal wellbores, precisely predicting flow patterns using conventional approaches is often problematic. Consequently, accurate monitoring and analysis of water holdup in gas–water two-phase flows are essential. This study performs a gas–water two-phase flow simulation experiment under diverse total flow and water cut conditions, utilizing air and tap water to represent downhole gas and formation water, respectively. The experiment relies on the measurement principles of an array fiber holdup meter (GAT) and the response characteristics of the sensors. In the experiment, GAT was utilized for real-time water holdup measurement, and the acquired sensor data were analyzed using three interpolation algorithms: simple linear interpolation, inverse distance weighted interpolation, and Gaussian radial basis function interpolation. The results were subsequently post-processed and visualized with 2020 version MATLAB software, generating two-dimensional representations of water holdup in the wellbore. The study findings demonstrate that, at total flow of 300 m^3^/d and 500 m^3^/d, the simple linear interpolation approach yields superior accuracy in water holdup calculations, with imaging outcomes closely aligning with the actual gas–water flow patterns and the authentic gas–water distribution. As total flow and water cut increase, the gas–water two-phase flow progressively shifts from stratified smooth flow to stratified wavy flow. In this paper, the Gaussian radial basis function and inverse distance weighted interpolation algorithms exhibit superior accuracy in water holdup calculations, effectively representing the fluctuating features of the gas–water interface and yielding imaging outcomes that align more closely with experimentally observed gas–water flow patterns.

## 1. Introduction

Water holdup is a critical flow parameter in production logging for oil well profile, as it immediately indicates the distribution of the water phase in oil and gas wells. Measuring water holdup enables production personnel to accurately locate water-producing zones, evaluate the well’s production status, and refine production methods [1,2]. Water holdup measurement is an essential technique for production assessment, and it is crucial for maintaining steady and high-yield output, particularly in oil and gas wells. Gravity separation frequently transpires downhole owing to the disparate physical properties of the lighter phase (e.g., gas) and the heavier phase (e.g., water). The denser phase typically descends, whilst the less dense phase ascends, markedly affecting fluid distribution during wellbore flow. The distribution patterns of water retention vary between vertical and horizontal wells. In horizontal wells, gravity separation is more evident, resulting in more prominent fluid stratification features.

In contrast to vertical wells, horizontal wells demonstrate notable variations in wellbore morphology and architecture, complicating water holdup assessments. Traditional water holdup logging devices are often engineered for vertical wells, considering fluid distribution properties in the vertical axis. Nevertheless, the application of these logging devices in horizontal wells is hindered by the significant gravity separation of fluids, which complicates the ability of traditional instruments to effectively measure and represent the intricate flow patterns and water holdup distribution in such wells [3]. The precision and reliability of data are considerably compromised, resulting in mistakes in evaluation outcomes when employing typical water holdup instruments in horizontal wells.

The proliferation and utilization of horizontal wells are becoming prevalent in global oil and gas production, rendering the measurement of multiphase fluid water holdup in these wells a critical technical problem for production monitoring. International logging firms, notably Schlumberger, have achieved substantial progress in horizontal well measurement technology, creating an array of sophisticated integrated instruments for such measurements, including the Flagship, PS Platform, and Flo Scan Image systems [4,5,6,7,8]. The primary logging instruments utilized for measuring multiphase fluid water holdup in horizontal wells are the Capacitive Array Tool (CAT) and the Resistive Array Tool (RAT) [9,10,11,12,13,14,15]. These tools, functioning on distinct physical principles, can accurately measure the distribution of multiphase fluids, including gas, water, and oil, within the wellbore. The Capacitive Array Tool (CAT) quantifies the capacitive characteristics of fluids to precisely depict the water phase distribution in the wellbore, whereas the Resistive Array Tool (RAT) deduces fluid water holdup by assessing resistance fluctuations, rendering it particularly effective for high-resistivity oil and gas wells.

In recent years, numerous researchers have achieved substantial advancements in water-holdup measuring technologies and algorithm optimization, hence improving the accuracy and dependability of production monitoring in oil and gas wells through unique methodologies and instruments. Guo et al. [16] performed a comprehensive analysis of the response characteristics of three different water holdup measuring devices across diverse fluid flow conditions and suggested an enhanced selection framework for holdup parameters in oil–water two-phase flows within horizontal pipes. This research establishes a theoretical framework for selecting holdup measuring instruments and presents a scientific optimization approach for precisely determining holdup parameters in complex flow regimes, hence enhancing the accuracy and practicality of logging data. Song et al. [17] provided a computational method for determining phase holdup in CAT production logging, utilizing a differential approach based on cross-sectional subdivision of the wellbore. The approach attains accurate estimations of fluid distributions by meticulously subdividing the wellbore cross-section, with validation tests demonstrating an accuracy rate of 90%, signifying substantial practical utility. This novel methodology provides fresh perspectives on the measurement and analysis of complex fluids within the wellbore, which is especially significant in CAT production logging. Zhu et al. [18] developed and enhanced the Kriging technique from geographic statistics, integrating it with a multifactor fitting statistical model to provide a mathematical model for array holdup image processing applicable to various flow regimes. The enhanced Kriging technique more effectively accommodates the intricate distribution of multiphase fluids, while the multifactor fitting model increases the precision of logging data. This mathematical model serves as a powerful technical instrument for array holdup image processing, markedly enhancing the accuracy and resolution of logging imaging, hence rendering it particularly appropriate for production logging in intricate flow regimes and multiphase fluid environments. Shang et al. [19] employed optical fibers to generate fluorescence from mineral oil pollutants in water, enabling accurate quantification of oil concentration. Suehara et al. [20] developed a novel interface detector using plastic optical fiber to differentiate between air–oil and oil–water interfaces. The apparatus employs the variances in refractive indices among the three phases (air, oil, and water) to precisely identify surfaces. Meng et al. [21] performed a study on the efficacy of an in-line optical fiber analysis system tailored for crude oil surveillance in oil wells. Their research concentrated on assessing the system’s efficacy in quantifying critical characteristics, including oil composition and temperature, in the extreme conditions commonly encountered in oil wells. The system’s real-time monitoring capabilities were proven to be reliable and stable, delivering continuous data that could be crucial for optimizing oil extraction procedures. Wang et al. [22] conducted a study examining the quantification of gas holdup in oil–gas–water multiphase flows applying integrated conductance sensors. They designed and assessed a sensor system proficient in precisely detecting gas holdup by integrating several conductance sensors, hence enhancing measurement precision in intricate flow situations. Their findings indicated that the system can consistently acquire gas holdup data, which is essential for enhancing oil extraction and flow control in the petroleum sector.

This work employed an array fiber holdup meter (GAT) to perform gas–water two-phase flow experiments in a horizontal well under multiphase flow laboratory conditions. The gas–water flow patterns in the simulated wellbore were examined, and water holdup calculations were performed using three interpolation algorithms: simple linear interpolation, inverse distance weighted interpolation, and Gaussian radial basis function interpolation. The computed holdup values were juxtaposed with empirically obtained holdup data. The imaging outcomes of the three interpolation techniques were compared and analyzed.

This paper is structured such that Section 2 presents an overview of the gas–water two-phase flow simulation experiment. The variations in gas–water two-phase flow patterns were noticed by modifying various total flow and water cuts and elucidating the methodologies for measuring water holdup in gas–water two-phase flow across different total flow and water cuts. Section 3 analyses the operating principles of the array fiber holdup meter (GAT) and evaluates its response characteristics. Section 4 presents the fundamentals of simple interpolation, inverse distance weighted interpolation, and Gaussian radial basis function interpolation to calculate water holdup. Section 5 presents a comparative study of the experimental data, highlighting the findings of water holdup calculations and the two-dimensional imaging performance of three interpolation algorithms under various settings. The final section is the conclusion.

## 2. Gas–Water Two-Phase Flow Simulation Experiment

### 2.1. Experiment Overview

The experiment was performed in the high-angle horizontal well multiphase flow simulation laboratory at Yangtze University, as shown in Figure 1.

The experimental apparatus for gas–water two-phase flow comprises a simulated wellbore, a liquid storage tank, a gas storage tank, an air pump, a hydraulic system, and flow control devices for each phase, as seen in Figure 2. The mimicked wellbore consists of two glass wellbores with distinct characteristics. The wellbore measures a total length of 14.0 m, comprising 12.0 m of glass tubing with outer diameters of 139.7 mm and 177.8 mm, respectively. The simulated wellbore is affixed to an adjustable rotational pedestal capable of rotating from 0° to 90°, facilitating flexible modification of the wellbore inclination angle. The experiment examined the flow characteristics of gas–water two-phase flow by varying the inclination angle under varied well circumstances. An innovative three-stage gas–water separation tank system was utilized to guarantee the sustainability of the experiment and the reutilization of fluids. This apparatus efficiently segregated gas and water during the experiment. The processed gas was securely released into the atmosphere via the exhaust system, while the water was reclaimed and returned to the storage tank for utilization in future experiments. This fluid recycling system markedly lowered water waste throughout the experiment while promoting environmental sustainability and eco-friendliness.

The experiment was performed in a laboratory setting at 20 °C and 95.89 kPa, using air with a density of 1.205 kg/m^3^ and a viscosity of $1.81\times 10^{-2}$ mPa·s, alongside tap water with a density of 998 kg/m^3^ and a viscosity of 1.16 mPa·s, to replicate downhole gas and formation water, respectively. Table 1 delineates the precise parameters for the gas–water two-phase flow studies. The simulation was conducted at multiple flow rates and water holdup levels with the wellbore positioned at a 90° angle. A total of 135 sets of experimental data were collected during the tests (including repeated measurement data).

### 2.2. Gas Array Tool Introduction

The gas array tool (GAT), developed by SONDEX, is primarily used to measure gas–liquid multiphase flow in horizontal and near-horizontal wells. It has a length of 159.1 cm, a weight of 10.1 kg, and is designed to operate under high-temperature and high-pressure conditions, with a maximum temperature of 350 °F and pressure up to 103.4 MPa. The tool can be used in casings with a maximum diameter of 7 inches. The primary structure comprises a central body and six spring arms, each fitted with a high-precision optical sensor, as seen in Figure 3a. Upon extension of the arms, the optical sensors are uniformly arranged throughout the inner surface of the wellbore, creating a measurement ring that facilitates thorough and precise data acquisition, as depicted in Figure 3b. This architecture allows the GAT to precisely capture the gas–liquid holdup distribution within the wellbore under intricate well circumstances. The fundamental elements of the GAT, comprising the communication board, interface board, and optical sensors, collaborate to guarantee precise data collection and processing in intricate well settings. The communication board is essential to the system. It not only delivers accurate power to the sensors, assuring uninterrupted and reliable performance in challenging downhole conditions, but also oversees data transfer. The interface board operates like a router, managing data from the six optical sensors. In multiphase fluid measurements, the interface board consolidates and processes data from optical sensors and employs a MEMS accelerometer to capture and document wellbore-related information.

The optical sensors, as integral components of the GAT, play a crucial role in precisely distinguishing the physical properties of various fluids within the wellbore. These sensors, owing to their heightened sensitivity, can promptly identify minute variations in the fluid, thereby ensuring the precision and dependability of the measurement data. Each of the six optical sensors differentiates gas from liquid by measuring light reflected from a conical sapphire surface, leveraging refractive index differences. With the sapphire refractive index set at 1.76, the surrounding medium’s refractive index helps determine the fluid phase: if the external medium is gas, the refractive index is approximately 1.0, whereas for liquid, it ranges between 1.33 and 1.55, as shown in Figure 4. This precise differentiation between gas and water enhances the accuracy of measurement outcomes, ensuring reliable data collection in complex downhole environments.

### 2.3. Analysis of Gas–Water Two-Phase Flow Pattern

In horizontal wells, the lighter gas phase generally resides above the water phase at low flow rates. Nevertheless, the gas–water two-phase flow patterns can become exceedingly intricate and varied due to the impact of parameters such as well inclination, phase flow rates, and fluid viscosity. Researchers, both nationally and globally, have categorized gas–water two-phase flow into six principal flow patterns: stratified smooth flow (SS), stratified wavy flow (SW), annular flow (A), elongated bubble flow (EB), slug flow (SL), and scattered bubble flow (DB) [23,24], as illustrated in Figure 5. In the illustration, yellow denotes the gas phase, while blue signifies the water phase.

This study involved gas–water two-phase flow simulation studies conducted at a 90° well inclination (horizontal well) under varying total flow and water cuts. Throughout the experiment, fluid flow patterns within the wellbore were visually monitored, and a high-speed camera was employed to acquire and document photographs, as illustrated in Figure 6, Figure 7 and Figure 8.

Figure 6, Figure 7 and Figure 8 illustrate that at total flow rates of 300 m^3^/d and 500 m^3^/d, the gas–water two-phase flow within the wellbore mostly displayed a characteristic stratified flow pattern. In this flow configuration, the gas phase is often situated at the upper section of the wellbore, whereas the water phase resides in the lower section, exhibiting a smooth and distinctly identifiable interface between the two phases. This flow condition, marked by a uniform interface, enables the assessment of fluid distribution and the computation of water holdup. As the total flow increased to 700 m^3^/d, the stratified flow pattern continued under low water-holdup circumstances, characterized by a distinct gas–water interface. Nevertheless, the interface began to display oscillations. An additional increase in the water cut exacerbated the variations in the gas–water interface, resulting in diminished stability of the stratified flow. The interface between the gas and water phases exhibited more prominent undulations, and the flow pattern progressively shifted from stratified flow to stratified wavy flow. In the stratified wavy flow, the gas and water phases ceased to exhibit a straightforward layered configuration, instead establishing a wavy gas–water interface, with the wave amplitude escalating as the water cut and total flow grew.

### 2.4. Measurement and Calculation Method of Water Holdup in Simulated Wellbore

This experiment utilized a transparent glass wellbore, enabling researchers to visually study the flow patterns of the gas–water two-phase flow within the wellbore. A calibrated ruler was utilized to guarantee the precision of data collecting measurements. Prior to the commencement of the experiment, the researchers secured the calibrated ruler to the exterior wall of the simulated wellbore, ensuring the zero mark was accurately aligned with the apex of the wellbore. This approach sought to reduce data variation resulting from scale misalignment or measurement inaccuracies, hence enhancing the precision of scale calculations and ensuring the trustworthiness of experimental outcomes [21,22]. The experiment replicated the flow characteristics of gas–water two-phase flow under different total flow and water cut circumstances. Data recording was performed solely when the flow pattern had stabilized to guarantee the accuracy of the experimental results. The actual flow states were documented as pictures, and the associated calibrated ruler values were noted, as seen in Figure 9, to guarantee the comprehensiveness of the experimental data.

The outer diameter of the simulated wellbore is 177.8 mm, while the inner diameter measures 159.0 mm. The GAT specs indicate that the main body diameter is 43.0 mm. Figure 10 illustrates the cross-section of the simulated wellbore.

This experiment calculates the water holdup in the gas–water two-phase flow by measuring the sector area occupied by the water phase in the wellbore, as seen in Figure 11. The calibrated ruler can be utilized to ascertain the arc length associated with the water phase. The central angle associated with the arc length is subsequently determined using Equation (1), after which the cross-sectional area occupied by the water phase is computed using Equation (2). The water holdup is ultimately calculated using Equation (3).
(1)$$\theta =\frac{l}{c}\times 360$$
(2)$$S_{w}=\frac{\theta }{360{}^{\circ}}\times S-\frac{1}{2}r^{2}\sin\theta$$
(3)$$Y_{w}=\frac{S_{w}}{S}$$

## 3. Response Characteristics Analysis of GAT

The GAT sensors were evenly spaced around the wellbore cross-section, enabling each sensor to autonomously gather fluid data from various locations. The non-uniform fluid distribution within the wellbore resulted in disparate response values from sensors positioned at various locations. In the single-phase calibration experiment, the GAT sensors were evaluated in circumstances of pure gas and pure water. The experimental results, illustrated in Figure 12, demonstrate notable variations in the GAT’s response values across different fluids. In a pure gas environment, the GAT’s response value approaches 1, indicating the sensor’s exceptional sensitivity to gas and its capacity to precisely characterize the gas phase. In a clean water environment, the GAT response value is roughly 0.2. Water possesses a higher refractive index than gas, leading to a markedly diminished sensor response. Due to the significantly larger response value for gas compared to water, the GAT can swiftly and reliably identify the presence of gas and precisely quantify its distribution within the fluid phase. This feature renders the GAT exceptionally adept at quantifying gas–water two-phase flows, particularly when the gas phase predominates in the fluid composition, hence enhancing measurement accuracy and dependability. The GAT is widely used in multiphase flow measurements in oil and gas wells due to its distinctive capability to detect gas, offering high-precision data for production monitoring, especially in intricate gas–water two-phase flow scenarios.

In this experiment, the data collected by the six sensors of the GAT were normalized using the following formula:(4)$$Y_{iw}=\frac{Y_{i}-Y_{ig}}{Y_{iw}-Y_{ig}}$$
where $Y_{iw}$ denotes the water holdup measured by the $i^{th}$ sensor, $Y_{i}$ signifies the measured value of the $i^{th}$ sensor, $Y_{ig}$ is the calibration value of the $i^{th}$ sensor in gas, and $Y_{iw}$ is the calibration value of the $i^{th}$ sensor in water; i=1,2,⋯,6.

## 4. Gas–Water Two-Phase Flow Imaging Algorithm of GAT

The imaging algorithm reconstructs the GAT response data to produce a two-dimensional image of the wellbore, enabling the investigation and identification of gas–water two-phase flow characteristics in horizontal wells. This provides the rapid evaluation of fluid flow patterns and interfacial properties. The imaging technique offers an accurate foundation for estimating water holdup, hence improving the comprehension of fluid flow dynamics and providing essential insights for well condition assessment and fluid management during actual production processes.

### 4.1. Simple Linear Interpolation Algorithm

The simple linear interpolation (SLI) algorithm is a widely utilized method for estimating unknown values between established data points. The fundamental concept is to establish a linear connection between two known data points and determine the values of intermediate points based on this line. The equation is presented in Equation (5).
(5)$$y=y_{0}+(x-x_{0})\times \frac{y_{1}-y_{0}}{x_{1}-x_{0}}$$
where $\left(x_{0},\;y_{0}\right)$ and $\left(x_{1},\;y_{1}\right)$ are the known points, and (x, y) is the unknown point.

According to the principle of the simple linear interpolation method, the wellbore cross-section is represented as a two-dimensional coordinate system, with each sensor assigned to a distinct coordinate. The water holdup at every location within the cross-section can be calculated through Equation (6).
(6)$$Y_{w}\left(x\right)=Y_{iw}+(x-x_{i})\frac{Y_{jw}-Y_{iw}}{x_{j}-x_{i}}$$
where ($x_{i}$, $Y_{iw}$) and ($x_{j}$, $Y_{jw}$) denote the positional coordinates of the sensors inside a two-dimensional coordinate system, and $\left(x,\;Y_{w}(x\right)$) represents the coordinates of the location at which the water holdup is to be calculated in the same two-dimensional coordinate system.

### 4.2. Inverse Distance Weighted Interpolation Algorithm

The inverse distance weighted (IDW) interpolation algorithm is a widely employed interpolation method. The fundamental principle is predicated on the assumption of a certain relationship between measurement points with known data and target points with unknown data, wherein this relationship is inversely proportional to the distance between the two points. The proximity of the known point to the target point directly correlates with its influence; the nearer the point, the higher the influence, and conversely, the farther the distance, the lesser the influence. This relationship enables the use of information from established measurement places to infer values in unmeasured areas, rendering it appropriate for situations necessitating spatial data interpolation [25].

In the two-dimensional coordinate system of the circular cross-section of the wellbore, a point can be denoted as $P_{i}(x_{i},\;Y_{j})$. The water holdup at a certain location can be determined using the GAT water holdup response data, as illustrated in Equation (7).
(7)$$Y_{wp}=\sum_{i=1}^{6}D_{ij}*Y_{wi}$$
where $Y_{wp}$ denotes the water holdup at a specific point in the wellbore, $D_{ij}$ is the distance weight value from the $i^{th}$ sensor to the target point j in the wellbore cross-sectional coordinate system, $Y_{wi}$ is the water holdup response value of the $i^{th}$ sensor, and i, j=1,2,⋯,6.

The inverse distance weighted interpolation algorithm employs the Euclidean distance equation, with the calculation formula for $D_{ij}$ specified in Equation (8).
(8)$$D_{ij}=\frac{1}{\sqrt{\sum_{i}{\left(x_{i}-y_{i}\right)}^{2}}}$$

Employing the inverse distance weighted interpolation algorithm to calculate the fluid water holdup in the wellbore, the response values of the six GAT probes are denoted as $Y_{w1},Y_{w2},\;\cdots Y_{w6}$, with corresponding coordinates $\left(x_{1},\;y_{1}\right),\;\left(x_{2},\;y_{2}\right),\;\cdots ,\;(x_{6},\;y_{6})$. Equation (9) provides the formula for calculating the water holdup in the wellbore.
(9)$$Y_{w}=\frac{\sum_{i=1}^{6}{D_{ij}}^{2}Y_{wi}}{\sum_{i=1}^{6}{D_{ij}}^{2}}$$
where $Y_{w}$ denotes the water holdup of the fluid within the wellbore, $D_{ij}$ is the distance weight value from the $i^{th}$ sensor to the target point j in the wellbore cross-sectional coordinate system, $Y_{wi}$ is the water holdup response value of the $i^{th}$ sensor, and i, j=1,2,⋯,6.

### 4.3. Gaussian Radial Basis Function Interpolation Algorithm

The radial basis function (RBF) is a real-valued function that calculates values according to the distance from a central point, commonly employed in interpolation, function approximation, and regression tasks. The value of the RBF is contingent upon the variation in distance from a point to the origin, with the function value modifying in accordance with changes in distance. The RBF value diminishes with increasing distance and increases with decreasing distance. The RBF is extremely efficient in addressing computational issues related to geographical distribution [26]. The fundamental structure of a radial basis function calculates its result based on the distance between the independent variables. When the distance from a specified point to the origin is considered the independent variable, the radial basis function may be articulated using the following formula:(10)$$\emptyset \left(r\right)=f(\left\|x-x_{0}\right\|)$$
where r denotes the distance from the known point to the origin, x is the target location, and $x_{0}$ is the known point. The distance serves as the fundamental input parameter of the radial basis function, dictating the output value of the function.

The Gaussian radial basis function (GRBF) is a prevalent interpolation method within the RBF family that is applied extensively across diverse domains. Its robust numerical approximation abilities facilitate precise calculation throughout a comprehensive domain. It delivers precise numerical data for any location within the cross-sectional space, regardless of whether it is a target point or a known point. The formula for the interpolation algorithm of the Gaussian radial basis function (RBF) is as follows:(11)$$\varphi (r)=e^{\frac{-r^{2}}{\gamma^{2}}}$$

Equation (12) presents the formula to calculate the distance weighting factor between two positions.
(12)$$D_{ij}=e^{\frac{-[{\left(x_{i}-x_{j}\right)}^{2}+{\left(y_{i}-y_{j}\right)}^{2}]}{\gamma_{ij}^{2}}}$$
where $D_{ij}$ denotes the distance weighting factor between the known information point and the target point, and $\gamma_{ij}$ is the decay control coefficient between the two locations, which reflects the fluid flow characteristics within the wellbore.

By calculating the distance weighting factor between two sample locations through the Gaussian radial basis function interpolation method, the water holdup at the specified location on the wellbore cross-section can be ascertained using the water holdup definition formula. The total water holdup of the gas–water two-phase flow in the horizontal well is then possible to calculate using the response values from the six GAT probes, as demonstrated in Equations (13) and (14).
(13)$$Y_{wp}=\sum_{i=1}^{6}D_{ij}*k_{i}*Y_{wi}$$
(14)$$Y_{w}=\frac{\sum_{i=1}^{6}{D_{ij}}^{2}{k_{i}Y}_{wi}}{\sum_{i=1}^{6}{D_{ij}}^{2}}=\frac{{D_{1}}^{2}k_{1}Y_{w1}+{D_{2}}^{2}{k_{2}Y}_{w2}+\ldots +{D_{6}}^{2}k_{6}Y_{w6}}{{D_{1}}^{2}+{D_{2}}^{2}+\ldots +{D_{6}}^{2}}$$
where $Y_{wp}$ denotes the water holdup at a specific location within the wellbore, $D_{ij}$ is the distance weight value from the $i^{th}$ sensor to the target point j in the wellbore cross-sectional coordinate system, $k_{i}$ is a specific coefficient that varies among different probes to maintain algorithm compatibility, $Y_{wi}$ indicates the water holdup response value of the $i^{th}$ GAT sensor, and i, j=1,2,⋯,6.

## 5. Results and Discussion

### 5.1. Comparison of Water Holdup Calculation Results Using Different Imaging Algorithms

In the experiment, the local water holdup response data of GAT sensors under different conditions were collected and organized, including repeated measurements to ensure the accuracy and reliability of the data. The experimental conditions included total flows of 300, 500, and 700 $m^{3}/d$, as well as water cuts of 15%, 30%, 50%, and 80%, as seen in Table 2, Table 3 and Table 4.

Three interpolation methods were employed to calculate the water holdup, and the results were meticulously compared, highlighting the applicability and performance variances of each algorithm over diverse flow patterns and water cut circumstances, as illustrated in Table 5, Table 6 and Table 7.

Table 5, Table 6 and Table 7 demonstrate that at total flows of 300 m^3^/d and 500 m^3^/d, the gas–water two-phase flow within the wellbore displayed a characteristic stratified flow, characterized by a distinct and smooth interface between the gas and water phases. The water holdup determined by the simple linear interpolation approach was the most accurate compared to the experimental readings. Due to the generally smooth and linear nature of the interface in stratified smooth flow, the simple linear interpolation approach successfully represented the gas–water interface characteristics, yielding precise water holdup results. This method exhibited consistent computation accuracy under both low and high water-holdup scenarios.

The water holdup calculated using the inverse distance weighted interpolation algorithm was markedly beyond the experimental values, exhibiting a considerably greater inaccuracy. As the water holdup worsened, this mistake became increasingly evident, especially under conditions of elevated water holdup, where the algorithm failed to effectively delineate the gas–water boundary, leading to greater discrepancies in the estimated water holdup.

The Gaussian radial basis function interpolation algorithm, although marginally less precise than the basic linear interpolation approach, yielded findings that were nevertheless relatively near the experimental data. Through seamless interpolation, it managed the gas–water distribution in the wellbore effectively, but with some flaws in boundary detection. This algorithm provided consistent and precise water holdup calculations.

Figure 13, Figure 14 and Figure 15 illustrate the relative errors between the calculated water holdup and the experimentally measured values for each of the three interpolation algorithms, hence evaluating the accuracy of the calculated water holdup in comparison to the actual measurements.

The following conclusions can be derived from Figure 13, Figure 14 and Figure 15:(1)In stratified flow within the simulated wellbore, the simple linear interpolation approach exhibits great accuracy, especially under medium- and low-flow circumstances, with a relative error not exceeding 10%. The uncomplicated linear interpolation approach accurately represents the ratio of the gas and water phases owing to the distinct and unobstructed gas–water interface in stratified flow. Nonetheless, as the flow pattern shifts from stratified flow to stratified wavy flow, the relative error in the water holdup estimate markedly escalates due to the oscillating interface. The algorithm fails to accurately reflect the actual distribution of the gas and water phases;(2)The inverse distance weighted interpolation algorithm demonstrated suboptimal performance across all three flow rate circumstances, resulting in the highest total inaccuracy. Under stratified smooth flow conditions, the algorithm was unable to accurately represent the flat properties of the gas–water interface, resulting in considerable discrepancies in the calculated results. Nonetheless, as the flow rate and water retention escalated, particularly during the transition to stratified wavy flow, the algorithm’s accuracy enhanced. The intricacy of stratified wavy flow enabled the algorithm to demonstrate advantages in managing nonlinear interfaces, with the calculated outcomes progressively aligning with the experimental observations;(3)The Gaussian radial basis function interpolation algorithm approach yielded consistent results in medium- and low-flow circumstances, with a maximum overall error of 9.2%. Despite its marginally lower performance compared to the simple linear interpolation algorithm, it effectively represented the gas–water distribution, with calculated outcomes nearly aligning with the experimental data. As the water holdup or flow rate increased, especially when the total flow attained 700 m^3^/d, resulting in a transition from stratified smooth flow to stratified wavy flow, this algorithm exhibited commendable performance under these intricate conditions, yielding results that closely matched the experimental measurements.

### 5.2. Comparison of Imaging Effects by Different Imaging Algorithms

In this study, three interpolation algorithm codes were written using 2020 version MATLAB, into which the water holdup data measured and calculated by the GAT sensors were input, and finally an imaging map of the wellbore cross-section was generated.

Figure 16, Figure 17 and Figure 18 illustrate the water holdup imaging results from the GAT across varying total flows and water cut circumstances in the simulated wellbore. The imaging results derive from the GAT sensor response data for gas–water two-phase flow, and the two-dimensional water holdup distribution maps were generated applying three different interpolation algorithms.

The imaging results for the three interpolation algorithms were evaluated under different total flow and water cut conditions when the instrument was absent from the wellbore, as illustrated in Figure 19, Figure 20 and Figure 21.

Figure 16 and Figure 19 illustrate that, at a total flow of 300 m^3^/d and varying water cut circumstances, the gas–water two-phase flow pattern in the simulated wellbore was analyzed applying three different interpolation algorithms. The gas–water two-phase flow in the wellbore displayed clear stratified smooth flow characteristics, with the gas phase situated in the upper section and the water phase located in the lower section. The substantial density disparity between the gas and water phases resulted in a smooth and distinct interface. As the water retention progressively escalated, the gas–water contact remained comparatively constant. The simple linear interpolation approach yielded image results that most closely resembled the actual flow pattern seen in the experiment, precisely depicting the gas–water interface and mirroring the configuration of stratified smooth flow. The Gaussian radial basis function interpolation algorithm approach yielded excellent imaging results, largely conforming to the experimental flow pattern, albeit with some discrepancies. Nonetheless, it nonetheless precisely represented the gas–water contact in its entirety.

The imaging results from the inverse distance weighted interpolation approach exhibited considerable disparities when compared to the experimental flow pattern, particularly in depicting the water phase interface. The method displayed significant inaccuracies owing to the strong curvature of the water phase contact in the center of the wellbore cross-section, rendering it incapable of accurately representing the true stratified flow pattern. This inaccuracy resulted in diminished precision in characterizing stratified smooth flow, especially when the gas–water interface was smooth, where its imaging performance was markedly inferior to that of the other two interpolation algorithms.

According to Figure 17 and Figure 20, at a flow rate of 500 m^3^/d, the simple linear interpolation algorithm demonstrates outstanding performance, yielding images nearly indistinguishable from the experimental flow pattern. The Gaussian radial basis function interpolation algorithm yielded results that closely align with the actual flow pattern; however, its accuracy is marginally inferior to that of the simple linear interpolation algorithm, especially under medium-to-high water cut conditions, where minor discrepancies occur in the gas–water interface. The inverse distance weighted interpolation approach exhibited suboptimal performance, as the gas–water contact displayed curvature and a lack of smoothness. Consequently, its imaging outcomes failed to accurately represent the gas–water distribution, and the overall imaging quality was distinctly inferior to that of the other two interpolation algorithms.

Figure 18 and Figure 21 indicate that an increase in the total flow to 700 m^3^/d results in substantial alterations in the gas–water two-phase flow within the wellbore. In conditions of low water cut, the flow pattern within the wellbore exhibits a characteristic stratified smooth flow, with a distinct separation between the gas and water phases and a reasonably linear interface. As the water cut increases, the flow pattern shifts from stratified flow to stratified wavy flow. The gas–water interface exhibits a wavy configuration instead of a linear one, attributed to the interaction between fluids and the increased flow velocity, which complicates the flow pattern. Under low water cut conditions, the simple linear interpolation approach effectively represents the actual flow pattern, with imaging results aligning with the observed stratified smooth flow. Nonetheless, when water cut increases, the efficacy of the simple linear interpolation approach deteriorates markedly under medium and high water cut conditions. It inadequately represents the intricate stratified wavy flow, resulting in imaging outcomes that do not correspond to the actual flow pattern. Under conditions of low water cut, the inverse distance weighted interpolation algorithm exhibits suboptimal performance and inadequately represents the stratified smooth flow. Nonetheless, as the water cut increases, the algorithm exhibits improved performance under medium and high water cut conditions, effectively capturing the undulating properties of the gas–water interface and yielding more precise imaging of the stratified wavy flow. The Gaussian radial basis function interpolation algorithm approach exhibits efficacy under low water cut conditions, precisely representing the stratified smooth flow within the wellbore, with imaging outcomes aligning with the actual flow pattern. Under medium and high water cut conditions, the Gaussian RBF interpolation algorithm maintains effective performance in capturing the wavy stratified smooth flow characteristics, albeit with significantly reduced accuracy compared to the inverse distance weighted interpolation algorithm.

## 6. Conclusions

The increasing utilization of horizontal well technology in oil and gas field development has rendered the examination of multiphase fluid flow characteristics in horizontal wells increasingly significant. The fluid flow patterns in horizontal wells are intricate and varied, particularly under gas–water two-phase flow conditions, where the distribution and interface morphology of the gas and water phases are crucial for interpreting production logging. Consequently, precise computation of water holdup and examination of flow patterns are essential components of production monitoring in horizontal wells.

The quantification of water holdup relies on a precise evaluation of fluid distribution within the wellbore, whilst the analysis of flow patterns provides the theoretical basis for this measurement and its interpretation. This work included carrying out a gas–water two-phase flow experiment through an array fiber water holdup logging device (GAT). The experiment replicated gas–water two-phase flow in a horizontal well, measuring and analyzing the gas–water contact within the wellbore using the GAT data collecting system.

To evaluate the distribution of water holdup and the peculiarities of flow patterns, simple linear interpolation, inverse distance weighted, and Gaussian radial basis function interpolation algorithms were used to calculate water holdup from the GAT experimental data. The calculated outcomes were compared with the experimentally collected water holdup, and the imaging effects of the three interpolation algorithms across various flow patterns were analyzed and compared. Subsequent conclusions were drawn as follows:(1)The water holdup calculation results from the three interpolation algorithms indicate that, at total gas–water two-phase flows of 300 m^3^/d and 500 m^3^/d under varying water cut conditions, the simple linear interpolation algorithm exhibits high accuracy, with relative errors remaining below 10%. This algorithm accurately quantifies the ratio of gas to water phases. The inverse distance weighted interpolation algorithm approach inadequately represents the distribution of gas and water phases, leading to substantial discrepancies, with errors surpassing 10%. The Gaussian radial basis function interpolation algorithm, although marginally less precise than the linear interpolation algorithm, still yields reasonably accurate outcomes, with its computations closely aligning with experimental data. As the total flow increases to 700 m^3^/d, both the simple linear interpolation approach and the Gaussian radial basis function interpolation algorithm consistently yield water holdup results that approximate the experimental values under low water cut conditions. Nevertheless, as the water cut increases, the relative error of the simple linear interpolation algorithm approach increases markedly, rendering it incapable of accurately representing the actual distribution of the gas and water phases. The inverse distance weighted interpolation algorithm and the Gaussian RBF interpolation algorithm exhibit superior accuracy, as their outcomes align more closely with the experimental data;(2)The imaging results of the three interpolation algorithms indicate that at total gas–water two-phase flows of 300 m^3^/d and 500 m^3^/d, under varying water cut conditions, the wellbore flow exhibits typical stratified smooth flow, with the gas phase situated in the upper section and the water phase in the lower section. The contact between the gas and liquid phases is distinct and well defined. The simple linear interpolation algorithm approach yields optimal imaging results in this flow pattern, closely aligning with the experimentally observed flow pattern. It precisely delineates the seamless interface between the gas and liquid phases. The inverse distance weighted interpolation algorithm approach, however, inadequately represents the flat properties of the gas–water interface, particularly in relation to the water phase. The water phase interface in the imaging results is overly concave towards the center of the wellbore, failing to adequately represent the stratified smooth flow observed in the experiment, hence rendering its imaging results considerably worse than the other two algorithms. The Gaussian radial basis function interpolation algorithm approach, despite minor discrepancies, effectively represents the gas–water interface with commendable precision, closely approximating the experimental flow pattern. The algorithm’s smooth interpolation characteristic enables it to manage certain boundary details proficiently; yet, in instances with a flat interface, the simple linear interpolation algorithm has a tiny advantage. As the total flow increases to 700 m^3^/d, under conditions of low water cut, the flow pattern within the wellbore persists in a stratified smooth state, with the gas and water phases distinctly separated, and the gas–water interface staying linear and well defined. The simple linear interpolation algorithm approach accurately represents the flat properties of the gas–water interface, with the water holdup imaging results aligning with the stratified flow seen in the experiment. Nevertheless, the inverse distance weighted interpolation algorithm approach continues to inadequately represent the actual flow pattern in stratified smooth flow circumstances. The Gaussian radial basis function interpolation algorithm approach demonstrates constant efficacy, effectively representing the stratified flow distribution within the wellbore, with imaging outcomes aligning with the experimental flow pattern. While it is marginally less precise than the simple linear interpolation algorithm, its smooth interpolation feature renders it more robust in managing intricate boundaries. As water cut increases, the flow pattern within the wellbore progressively shifts from stratified smooth flow to stratified wavy flow, resulting in a wavy gas–water interface instead of a linear one. The simple linear interpolation algorithm approach fails to accurately represent the undulating features of the gas–water interface, leading to imaging outcomes that do not align with the actual flow pattern. The inverse distance weighted interpolation algorithm approach effectively captures the undulating characteristics of the gas–water interface, correctly representing the actual stratified wavy flow. The Gaussian radial basis function interpolation algorithm effectively operates in stratified wavy flow circumstances, accurately representing the undulating features of the gas–water interface. While it is marginally less effective in addressing specific boundary constraints than the inverse distance weighted interpolation algorithm, it nonetheless offers a generally accurate representation of the changing flow pattern.(3)The efficacy of the three interpolation algorithms for water holdup calculation accuracy and imaging outcomes differs across various total flow and water cut conditions. The study results demonstrate that in cases of stratified smooth flow patterns in gas–water two-phase flow, the simple linear interpolation algorithm is the most effective option. This approach precisely delineates the border between gas and water, yielding imaging results that closely align with actual data, and it also provides exceptional accuracy in water holdup calculations. Conversely, when the flow pattern shifts to stratified wavy flow, the Gaussian radial basis function interpolation algorithm and the inverse distance weighted interpolation algorithm are more effective in managing the nonlinear interface. Both yield more accurate water holdup estimations, and their imaging outcomes closely correspond with experimental findings.

While each interpolation algorithm has distinct advantages in varying circumstances, opportunities for enhancement persist. Given the escalating complexity of production well conditions in oil and gas field development, future research must prioritize the optimization of these algorithms to improve their stability and accuracy in more intricate fluid flow conditions. In practical applications for production wells, flow patterns are frequently more varied and complex, necessitating algorithms that can effectively adjust to swift alterations in flow patterns.

## Figures and Tables

**Figure 1 sensors-24-07285-f001:**
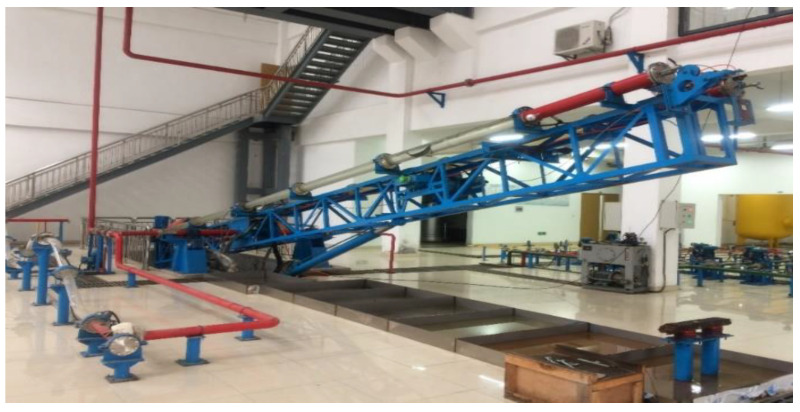
A real photo of the multiphase flow laboratory.

**Figure 2 sensors-24-07285-f002:**
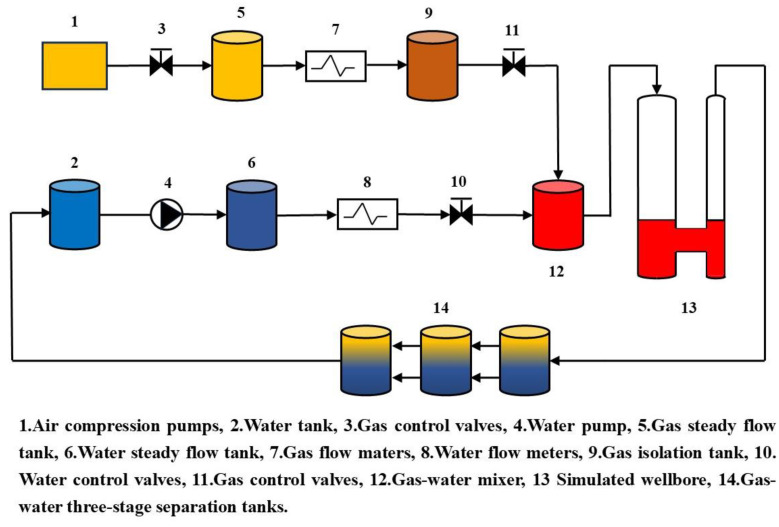
Schematic diagram of multiphase flow experiment device.

**Figure 3 sensors-24-07285-f003:**
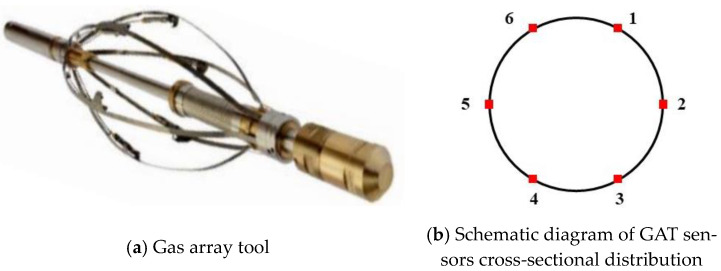
A physical photo of the gas array tool.

**Figure 4 sensors-24-07285-f004:**
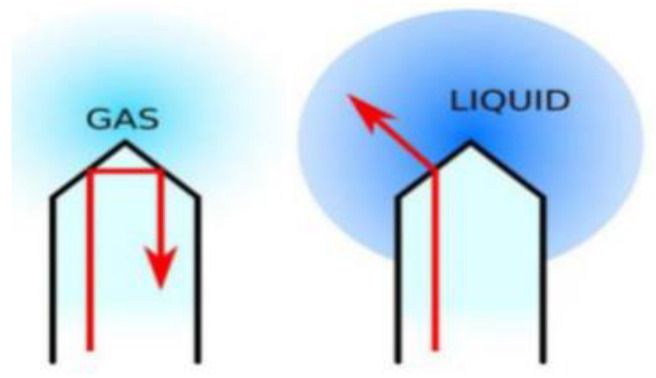
Schematic diagram of the light refraction principle of the GAT sensor.

**Figure 5 sensors-24-07285-f005:**
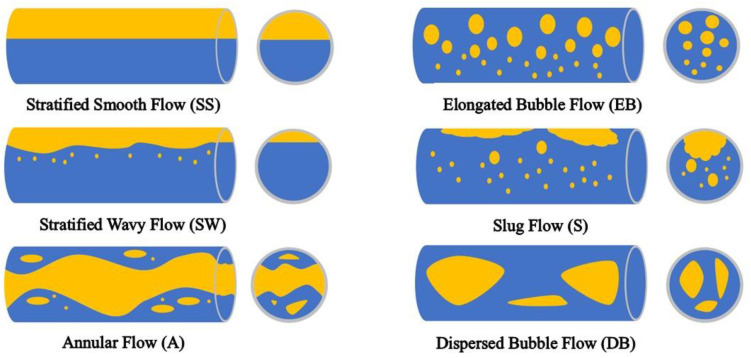
Flow pattern of two-phase gas–water flow.

**Figure 6 sensors-24-07285-f006:**
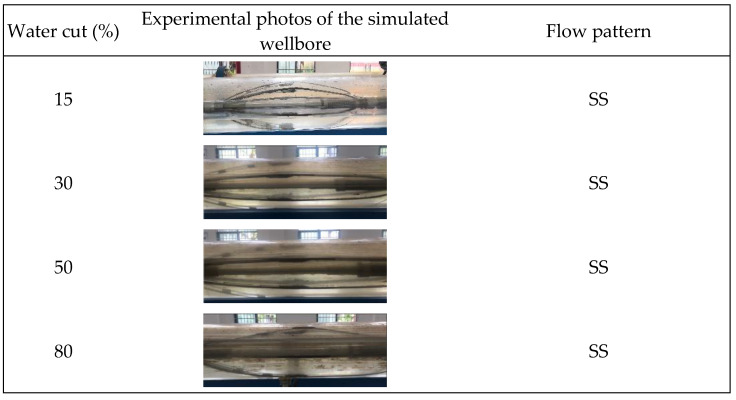
Experimental, physical photos of gas–water two-phase flow patterns under different water cuts when the total flow is 300 $m^{3}/d$.

**Figure 7 sensors-24-07285-f007:**
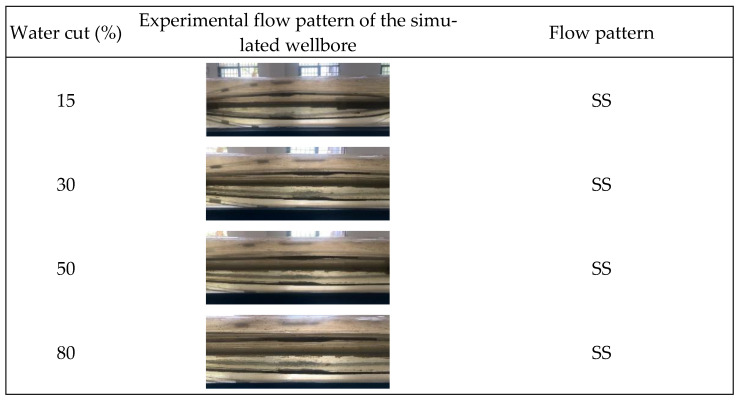
Experimental, physical photos of gas–water two-phase flow patterns under different water cuts when the total flow is 500 $m^{3}/d$.

**Figure 8 sensors-24-07285-f008:**
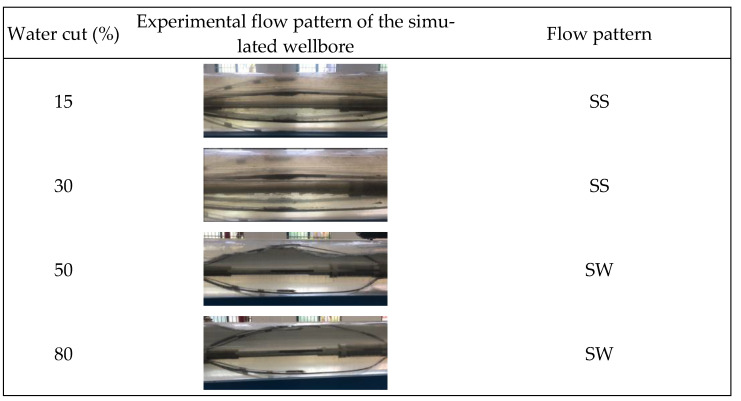
Experimental, physical photos of gas–water two-phase flow patterns under different water cuts when the total flow is 700 $m^{3}/d$.

**Figure 9 sensors-24-07285-f009:**
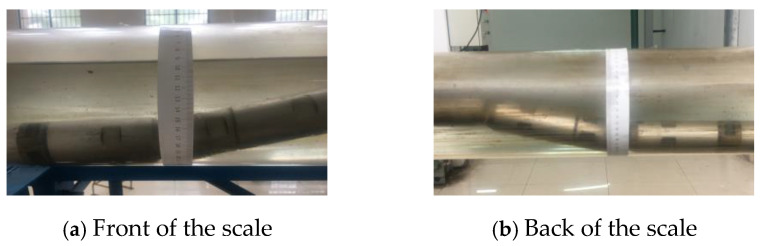
Physical photos of gas–water two-phase water holdup scale.

**Figure 10 sensors-24-07285-f010:**
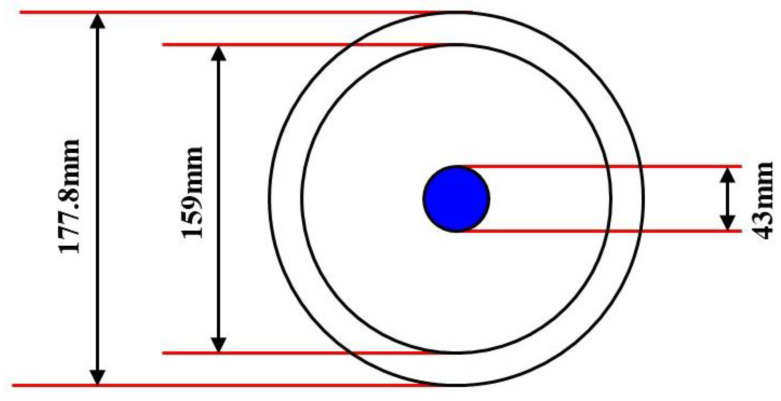
Schematic cross-section of wellbore.

**Figure 11 sensors-24-07285-f011:**
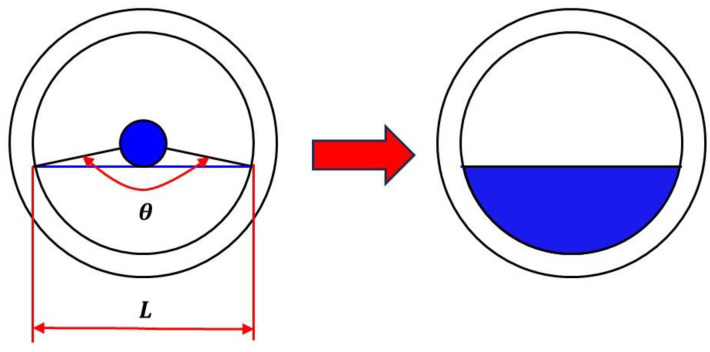
Flow chart of water holdup calculation.

**Figure 12 sensors-24-07285-f012:**
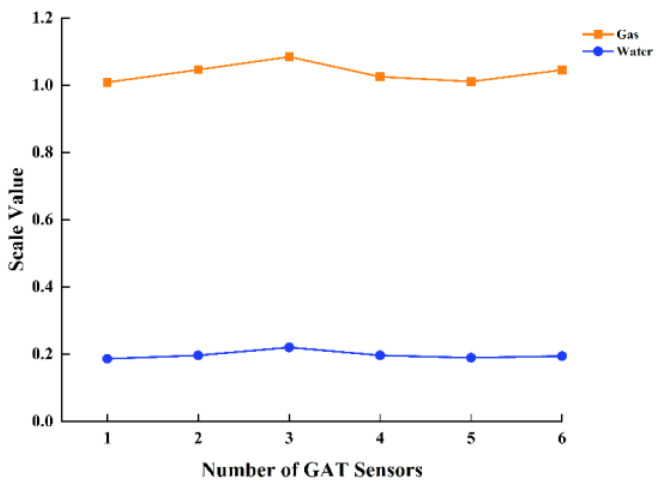
Single-phase scale cross-plot of GAT.

**Figure 13 sensors-24-07285-f013:**
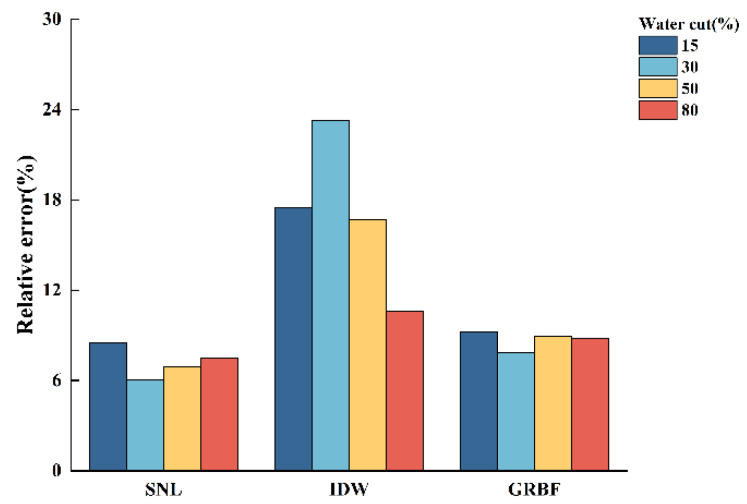
Comparison of the relative errors of three interpolation algorithms in calculating water holdup when the total flow is 300 $m^{3}/d$.

**Figure 14 sensors-24-07285-f014:**
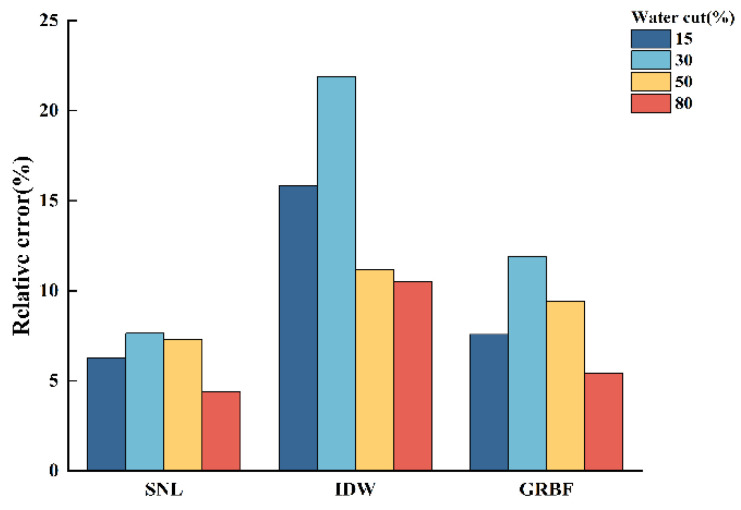
Comparison of the relative errors of three interpolation algorithms in calculating water holdup when the total flow is 500 $m^{3}/d$.

**Figure 15 sensors-24-07285-f015:**
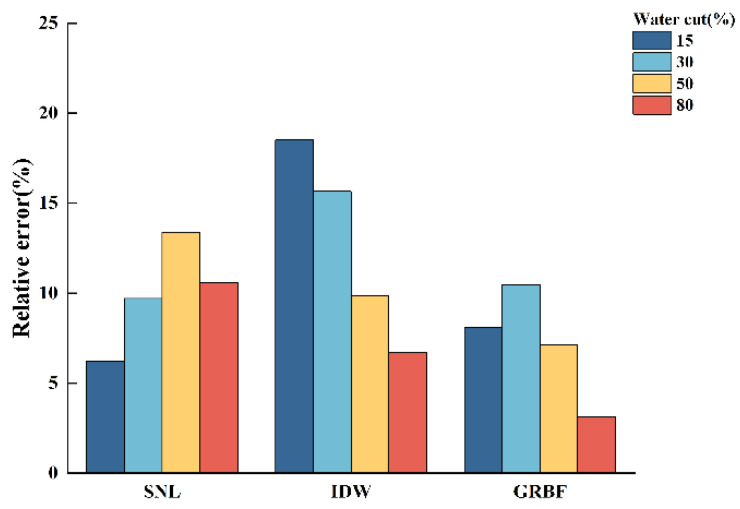
Comparison of the relative errors of three interpolation algorithms in calculating water holdup when the total flow is 700 $m^{3}/d$.

**Figure 16 sensors-24-07285-f016:**
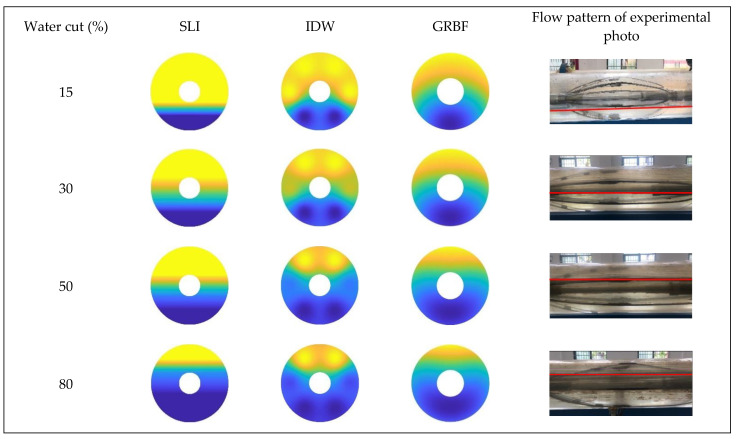
Comparison of imaging diagrams of three imaging algorithms and flow patterns of experimental photos when the total flow is 300$\;m^{3}/d$.

**Figure 17 sensors-24-07285-f017:**
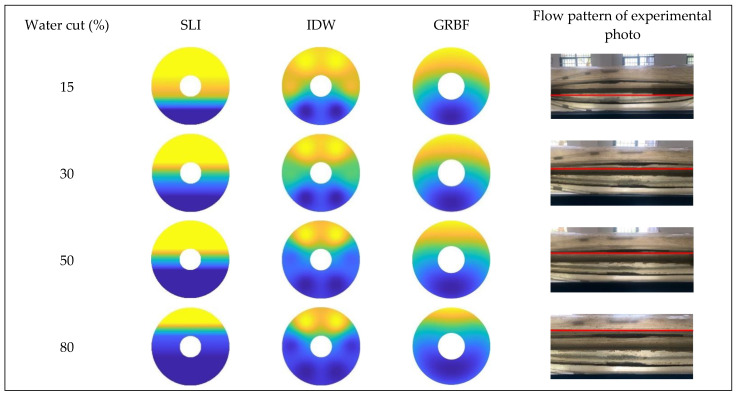
Comparison of imaging diagrams of three imaging algorithms and flow patterns of experimental photos when the total flow is 500 $m^{3}/d$.

**Figure 18 sensors-24-07285-f018:**
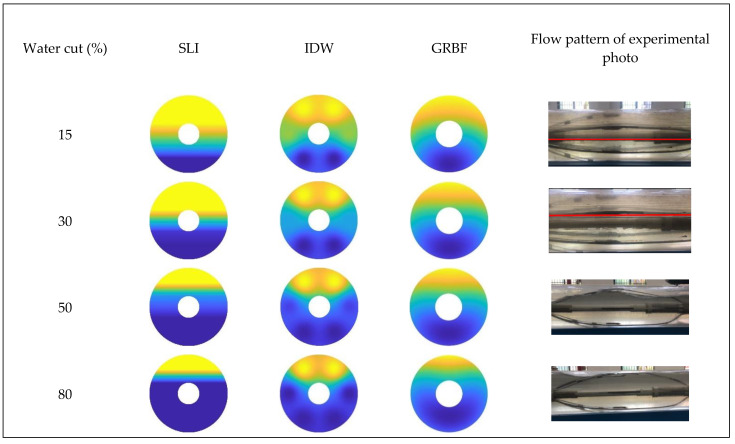
Comparison of imaging diagrams of three imaging algorithms and flow patterns of experimental photos when the total flow is 700 $m^{3}/d$.

**Figure 19 sensors-24-07285-f019:**
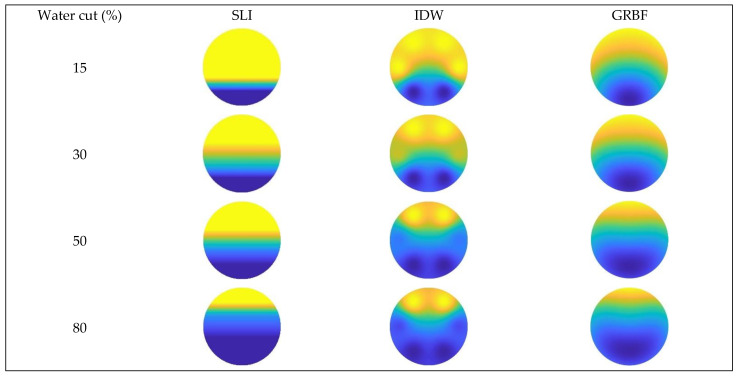
Comparison of imaging diagrams of three imaging algorithms without GAT when the total flow is 300 $m^{3}/d$.

**Figure 20 sensors-24-07285-f020:**
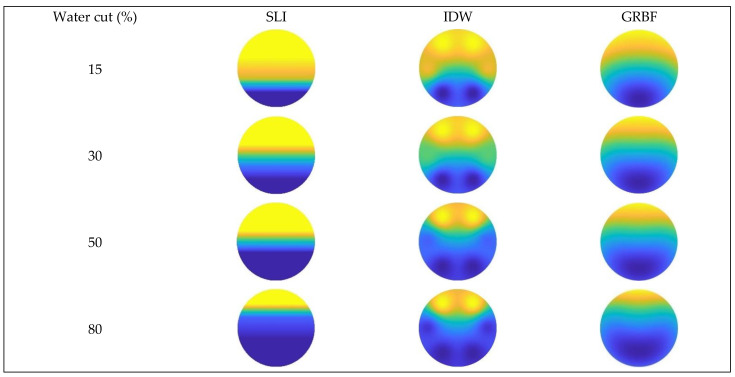
Comparison of imaging diagrams of three imaging algorithms without GAT when the total flow is 500 $m^{3}/d$.

**Figure 21 sensors-24-07285-f021:**
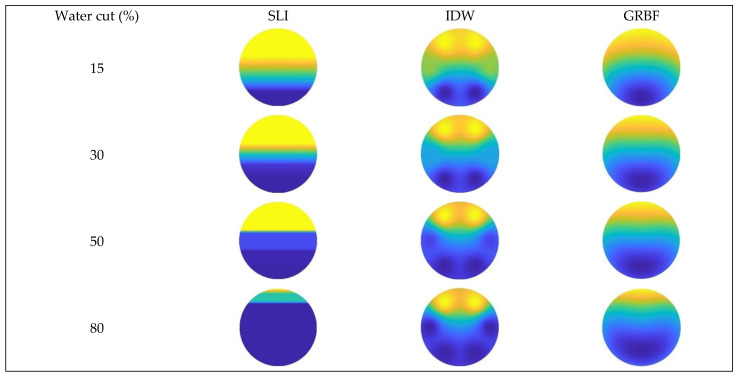
Comparison of imaging diagrams of three imaging algorithms without GAT when the total flow is 700 $m^{3}/d$.

**Table 1 sensors-24-07285-t001:** Schematic table of gas–water two-phase flow experimental design.

Well Deviation (°)	$\mathrm{Total}\;\mathrm{Flow}\;(m^{3}/d)$	Water Cut (%)
90	300, 500, 700	15, 30, 50, 80

**Table 2 sensors-24-07285-t002:** Water holdup response value of each GAT optical sensor when the total flow is 300 $m^{3}/d$.

Water Cut (%)	Number of GAT Optical Sensors					
	1	2	3	4	5	6
15	1	0.9981	0.2288	0.2122	1	1
30	1	0.8242	0.1945	0.1976	0.8179	1
50	1	0.6015	0.1956	0.1960	0.5935	1
80	0.9955	0.1996	0.1962	0.1951	0.1970	0.9892

**Table 3 sensors-24-07285-t003:** Water holdup response value of each GAT optical sensor when the total flow is 500 $m^{3}/d$.

Water Cut (%)	Number of GAT Sensors					
	1	2	3	4	5	6
15	1	0.9745	0.2122	0.2038	0.9718	1
30	1	0.7816	0.1999	0.2010	0.7532	1
50	1	0.5818	0.1977	0.1978	0.6031	1
80	0.9822	0.1975	0.1969	0.1955	0.1949	0.9860

**Table 4 sensors-24-07285-t004:** Water holdup response value of each GAT optical sensor when the total flow is 700 $m^{3}/d$.

Water Cut (%)	Number of GAT Sensors					
	1	2	3	4	5	6
15	1	0.9815	0.2052	0.2105	0.9888	1
30	1	0.8033	0.1983	0.2042	0.7955	1
50	1	0.5522	0.1975	0.1960	0.5491	1
80	0.9725	0.1976	0.1953	0.1965	0.1972	0.9665

**Table 5 sensors-24-07285-t005:** Comparison of water holdup calculated by different interpolation algorithms and experimental values when the total flow is 300 $m^{3}/d$.

Water Cut (%)	Experimental Value	SLI	IDW	GRBF
15	0.1534	0.1666	0.1805	0.1677
30	0.3166	0.3357	0.3925	0.3415
50	0.5292	0.5655	0.6173	0.5765
80	0.8353	0.8978	0.9239	0.9086

**Table 6 sensors-24-07285-t006:** Comparison of water holdup calculated by different interpolation algorithms and experimental values when the total flow is 500 $m^{3}/d$.

Water Cut (%)	Experimental Value	SLI	IDW	GRBF
15	0.1610	0.1712	0.1866	0.1733
30	0.3249	0.3496	0.3959	0.3635
50	0.5332	0.5719	0.5926	0.5832
80	0.8526	0.8898	0.9422	0.8988

**Table 7 sensors-24-07285-t007:** Comparison of water holdup calculated by different interpolation algorithms and experimental values when the total flow is 700 $m^{3}/d$.

Water Cut (%)	Experimental Value	SLI	IDW	GRBF
15	0.1655	0.1757	0.1959	0.1787
30	0.3299	0.3618	0.3814	0.3643
50	0.5303	0.6021	0.5827	0.5683
80	0.8680	0.9596	0.9259	0.8949

## Data Availability

The original contributions presented in this study are included in the article. Further inquiries can be directed to the corresponding author.

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
