# Peer review of "Application of Imaging Algorithms for Gas–Water Two-Phase Array Fiber Holdup Meters in Horizontal Wells"

_sensors, 2024, doi:10.3390/s24227285_

Round 1
Reviewer 1 Report
Comments and Suggestions for Authors
Dear Authors,
Please find all comments in the attached PDF file.
All the best

Author Response
We sincerely appreciate the valuable comments and questions raised by the reviewers. Your thoughtful insights and meticulous review have greatly contributed to enhancing the academic quality of this paper. We have carefully reviewed and analyzed your feedback and have made corresponding revisions and additions to the content. Here, we will provide detailed responses to each of your questions, hoping that our explanations can clearly address your concerns. Once again, thank you for your attention and support for our work.

Reviewer 2 Report
Comments and Suggestions for Authors
The manuscript investigates the application of three interpolation algorithms for water holdup imaging using a Gas Array Tool (GAT) in gas-water two-phase flow within horizontal wells. This study aims to improve measurement accuracy under varying flow and water cut conditions. Detailed comments are listed below:
1. Table 1 presents 12 unique test conditions at a 90° well deviation across three flow rates (300, 500, and 700 m³/d) and four water cuts (15%, 30%, 50%, and 80%). Could the authors clarify if the 135 datasets resulted from multiple repetitions per condition or if additional variables were assessed?
2. Consistency in sensor calibration is critical for accurate results. How was calibration drift addressed across these tests?
3. Considering adding the recently published experimental studies of air-water two-phase flow in inclined and horizontal wells.
- https://doi.org/10.3390/en14030578
4. Details on the manufacturers and specifications of key experimental devices like the Gas Array Tool (GAT), optical sensors, flow control mechanisms, and water holdup measurement tools are requested to validate the precision of the experimental setup.
5. Why were only the SLI, IDW, and GRBF interpolation algorithms chosen? Were other algorithms considered and excluded? If so, what was the rationale?
6. Considering the complex flow regimes in horizontal wells, how effectively do the imaging algorithms perform under transitional or mixed flow conditions?
7. Discuss how variations in temperature or pressure might influence the performance of the GAT’s optical sensors and the accuracy of imaging results.
8. Lastly, describe any potential refinements or advancements that could enhance algorithm performance under operational conditions.
Author Response
We sincerely appreciate the valuable comments and questions raised by the reviewers. Your thoughtful insights and meticulous review have greatly contributed to enhancing the academic quality of this paper. We have carefully reviewed and analyzed your feedback and have made corresponding revisions and additions to the content. Here, we will provide detailed responses to each of your questions, hoping that our explanations can clearly address your concerns. Once again, thank you for your attention and support for our work.
1.In this study, the 12 unique test conditions presented in Table 1 are based on combinations of three flow rates (300, 500, and 700 m³/d) and four water cuts (15%, 30%, 50%, and 80%) at a 90° well deviation. To ensure statistical significance and reliability of the experimental results, multiple repetitions were conducted under each condition, which resulted in a total of 135 datasets. Additionally, we assessed slight variations in fluid distribution patterns, especially focusing on the characteristics of gas-water interface fluctuations under different flow rates and water cuts. Therefore, the 135 datasets primarily resulted from multiple repetitions for each condition and further observations of flow pattern characteristics. Through these measures, we were able to capture the dynamic variations of gas-water two-phase flow with greater accuracy;
2.In this study, ensuring consistency in sensor calibration was essential to maintain accurate results. To address potential calibration drift across tests, We periodically tested sensors in single-phase gas and water environments to confirm that response values aligned with expected baselines (e.g., near 1 for gas and approximately 0.2 for water). Any deviation from these baseline readings prompted immediate recalibration;
3.To strengthen the study, we are considering incorporating recent experimental studies on air-water two-phase flow in inclined and horizontal wells. These studies could provide valuable insights into flow pattern transitions, interface behaviors, and the effects of inclination angles, helping us better contextualize and validate our findings;
4.To ensure clarity regarding the precision of our experimental setup, here are part of the details on the key experimental devices:
(1)The GAT used in our study was manufactured by Sondex, specifically designed for measuring gas-liquid multiphase flow in horizontal and near-horizontal wells. The tool consists of six high-precision optical sensors mounted on spring arms that extend to uniformly position the sensors along the inner surface of the wellbore. This arrangement ensures accurate measurement of the gas-liquid distribution across varying flow conditions;
(2)The optical sensors integrated into the GAT are fabricated from sapphire, chosen for its high optical stability and sensitivity. With a refractive index of 1.76, the sensors can differentiate between gas (refractive index 1.00) and water (refractive index 1.33), allowing precise detection of gas-water interfaces and accurate holdup measurements. The sensors were calibrated to standard baselines before each test to maintain consistency in data acquisition;
(3)Flow rates were controlled using flow regulators, which allowed precise adjustment of gas and water phases to replicate different downhole flow rates (300, 500, and 700 m³/d). These regulators ensured a stable flow rate within ±1% accuracy, providing reliable control over experimental conditions;
(4)The water holdup measurements were supported by a calibrated ruler and visual monitoring system, allowing us to capture accurate holdup values across the cross-sectional area of the wellbore. The transparent glass wellbore, combined with real-time video capture, enabled precise documentation of the gas-water interface, which was essential for validating sensor readings;
These specifications highlight the precision of our setup, ensuring that our data is both accurate and reproducible under the given experimental conditions.
5.The selection of Simple Linear Interpolation (SLI), Inverse Distance Weighted (IDW), and Gaussian Radial Basis Function (GRBF) interpolation algorithms was guided by their specific strengths in addressing the characteristics of gas-water two-phase flow imaging within the wellbore:
(1)SLI was selected for its simplicity and effectiveness in capturing stratified smooth flow, where the interface between gas and water phases is relatively linear. This algorithm provides reliable results in conditions with minimal interface complexity, making it suitable for cases with low to moderate water cut;
(2)IDW was chosen due to its capability to account for the spatial proximity of known data points, which is valuable in flow conditions where localized variations are more pronounced. This method helps capture curvature and wavy patterns in the gas-water interface that appear in higher flow rates and water cut conditions;
(3)IDW was chosen due to its capability to account for the spatial proximity of known data points, which is valuable in flow conditions where localized variations are more pronounced. This method helps capture curvature and wavy patterns in the gas-water interface that appear in higher flow rates and water cut conditions;
(4)Other interpolation algorithms, such as Kriging or spline-based methods, were considered but ultimately excluded due to their higher computational demands and longer processing times, which were less compatible with real-time data processing needs in our experimental setup. Additionally, Kriging, while highly accurate, is often more suitable for larger spatial datasets with extensive data points, which was not the case in our relatively constrained wellbore cross-section. The selected algorithms (SLI, IDW, and GRBF) offered an optimal balance of computational efficiency and accuracy for the flow patterns encountered in this study.
6.The performance of the selected imaging algorithms—Simple Linear Interpolation (SLI), Inverse Distance Weighted (IDW), and Gaussian Radial Basis Function (GRBF)—under transitional or mixed flow conditions in horizontal wells was evaluated to understand their adaptability to complex flow regimes:
(1)SLI performs well in stratified smooth flow, where the gas-water interface is relatively distinct and linear. However, under transitional or mixed flow conditions, such as when the flow transitions to stratified wavy or slug flow, SLI's accuracy diminishes. It struggles to capture non-linear interfaces and fluctuating boundaries due to its inherently linear assumptions, making it less effective in highly dynamic flow regimes;
(2)IDW offers moderate adaptability to mixed flow conditions. Its reliance on distance weighting helps it capture localized changes in gas-water distribution, making it more effective than SLI for transitional flows with slight interface undulations. However, IDW may still exhibit limitations in accurately depicting sharp changes or irregular patterns, as it can overemphasize nearby data points, leading to inaccuracies in areas with high curvature or rapid phase changes;
(3)GRBF demonstrates the best performance among the three algorithms under transitional and mixed flow regimes. Its ability to handle nonlinear, wavy interfaces allows it to adapt to the complex flow patterns seen in gas-water two-phase flow transitions. GRBF's smooth interpolation helps capture irregularities in the interface, such as those found in slug flow or stratified wavy conditions, providing imaging results that closely align with experimental observations;
In summary, while SLI and IDW have limitations under transitional flows, GRBF proves to be the most effective algorithm for handling complex, mixed flow conditions.
7.Variations in temperature and pressure can significantly impact the performance of the Gas Array Tool (GAT)'s optical sensors and, consequently, the accuracy of imaging results in gas-water two-phase flow conditions.
(1) Temperature effect: The GAT's optical sensors differentiate fluids based on refractive index, which is temperature-dependent. As temperature changes, the refractive indices of gas and water also shift, potentially altering the optical sensor's sensitivity and affecting its ability to accurately distinguish between phases. Higher temperatures can lower the refractive index contrast between gas and water, leading to reduced measurement sensitivity; Temperature variations can cause slight thermal expansion or contraction in the sensor materials, potentially affecting calibration stability. For example, sapphire, used in the sensors for its high optical stability, may experience minimal but relevant thermal changes. If uncorrected, these changes could introduce slight shifts in baseline readings, impacting holdup accuracy and the fidelity of imaging results.
(2)Pressure effects: Temperature variations can cause slight thermal expansion or contraction in the sensor materials, potentially affecting calibration stability. For example, sapphire, used in the sensors for its high optical stability, may experience minimal but relevant thermal changes. If uncorrected, these changes could introduce slight shifts in baseline readings, impacting holdup accuracy and the fidelity of imaging results; Pressure fluctuations may also affect sensor calibration stability. Although sapphire sensors are generally resilient to high pressures, repeated pressure cycles can still lead to calibration drift over time, especially in ultra-deep wells. Without regular recalibration, this drift can lead to cumulative errors, affecting the accuracy of water holdup measurements and, by extension, the imaging outcomes.
Due to safety considerations, the experimental device in this study cannot simulate the gas-water two-phase flow in horizontal wells under high temperature and high pressure conditions.
8.To improve the performance of imaging algorithms such as SLI, IDW, and GRBF under operational conditions, subsequent research will consider several improvements and enhancements:
(1)Adaptive Algorithm Selection: Implementing an adaptive framework that dynamically selects the most suitable algorithm based on real-time flow conditions could significantly improve accuracy. For instance, in stratified smooth flows, the system could favor SLI for simplicity and efficiency, while switching to GRBF or IDW under wavy or slug flow conditions. This adaptability could ensure optimal performance across different flow regimes without manual intervention;
(2)Machine Learning Integration: Machine learning (ML) models could be trained on historical flow and holdup data to recognize patterns and predict the most effective interpolation technique in real-time. ML could also assist in refining the output of existing algorithms, reducing errors introduced by complex flow transitions and offering a more accurate, hybrid imaging solution tailored to specific well conditions;
(3)High-Resolution Sensor Arrays: Using a higher density of sensors across the cross-section of the wellbore would provide a more detailed data set, which could improve algorithmic accuracy. This would reduce interpolation gaps and provide richer data, enabling the algorithms to generate a finer-resolution image that better captures complex fluid distributions and phase transitions.
Round 2
Reviewer 1 Report
Comments and Suggestions for Authors
Dear Authors,
You have introduced some necessary changes and clarified the issues reported in the review report.
Two issues remain unsolved: the major one is related to the collected datasets, to which you replied in point 7 of your response report. I have seen this matter of confidentiality raised frequently. I totally understand the concerns related to data publication. But this begs a question of how any research work can be verified if the data obtained and used to publish the work as results without showing the basis on which those results interpretation and conclusions are derived from.
Regretfully, without showing at least the categories of the datasets and their composition, the work will be not complete.
The second matter is still unsolved by your reply on point 3. The definition of tool response is vague and non-descriptive. What is the unit of that response? what does represent? to use only the word "response" and scale value is not satisfying.
Your efforts in introducing most of the corrections needed are highly appreciated. looking forward to seeing your work published.
All the best,
Reviewer
Author Response
Thank you very much for your thoughtful and constructive feedback. I am grateful for the opportunity to further clarify these points to ensure the research is as transparent and complete as possible.
1.Confidentiality of Collected Datasets
Thank you for your suggestion. After team discussions, we have decided to provide the GAT sensor response data under different water cut and total flow conditions in the Results and Discussion section. These data will help verify the authenticity and completeness of this research, thereby enhancing the transparency and reliability of the study.
2.Definition of Tool Response
The "tool response" in our study refers to the output readings from each GAT optical sensor, which are collected in unitless values ranging from 0 to 1. These values reflect the optical sensor’s detection of the gas or water phase in the wellbore based on refractive index differences. For instance:
Gas Phase: A response value approaching 1 indicates the presence of gas, given its refractive index is close to 1.0.
Water Phase: A response value closer to 0.2 indicates the presence of water, which has a higher refractive index range between 1.33 and 1.55.
These values, being unitless, are normalized measurements that indicate the fraction of the phase detected by each sensor.